# Has Secondary Science Education Become an Elite Product in Emerging Nations?—A Perspective of Sustainable Education in the Era of MDGs and SDGs

Gazi Mahabubul Alam [1,2]

1 College of Overseas Education, Chengdu University, Chengdu 610106, China; gazimalamb@yahoo.com
2 Department of Foundation of Education, Faculty of Educational Studies, University Putra Malaysia, Serdang 43300, Selangor, Malaysia

**Abstract:** Education is considered the single most important tool that supports the achievement of a nation's sustainable development. However, if a particular education program itself deprives students with a lower socioeconomic status (SES) to access it and subsequently restricts them from achieving a better performance, should such an education program be labelled as sustainable education, supporting the achievement of the SDGs (Sustainable Development Goals)? This question remains to be answered. Science education, which is also treated as an "international product", is the most essential component in education required to ensure sustainable national development. Consequently, science education should be a "right-based education program" that every "capable student", regardless of his/her SES, is able to obtain. This motive should ideally ensure the best practice mode of sustainable development in education. Keeping this view in mind, this research was conducted in an emerging nation, namely Bangladesh, to examine whether secondary science education has become an elite product and its consequential effect on sustainable education. A qualitative research method that adopts a descriptive analysis of secondary data was primarily used. The secondary data were collected from the public archive(s). Findings suggest that, mostly, students with a privileged SES can access science education programs. Moreover, these students perform well in major public examination(s). Primary data further collected by Focus Group Discussions (FGDs) summarise that science education is an international product. The following artificial perception has thus developed. To participate in such a program, a significant informal budget from parents' pockets is required in order to perform well. This is an obvious conflict with the spirit of sustainable education and SDGs. Hence, policy reform guidelines for decent practice are provided to resolve this misleading perception.

**Keywords:** sustainable education; SDGs; education and development; elitist view in education; socioeconomic status and education

## 1. Introduction

One of the central agendas of the sustainable development of education suggests that education quality should be unified [1]. Ideally, the same quality education should be provided to all children regardless of their SES, race, gender, and religious faith [2]. The agenda further recommends that the availability of subjects and programs should ideally be controlled according to the needs of the country, where students' abilities, merits, competencies, and interests would serve as the basis for allocating subjects and programs to be studied [3]. Providing prejudice based on SES, race, and other factors in order to study a favourable subject or a program is not just labelled as a form of discrimination and disparity, but it is considered a graver threat to sustainable development in education [4]. Although the effect of several issues (such as private schools, private tuition, parental interventions, and favourable policies) on the substantiable development of education deserves research attention, the impact of elitism in science education is an intense issue [5].

The writing that follows explains why this issue should be a substantial research problem that is to be studied more.

Education is generally viewed as the most powerful weapon for national development, as argued by Alam et al. [6]. Following their arguments, education is often considered a public good. However, debates are ongoing regarding whether all forms of education should be a part of public goods [7]. An enormous consensus suggests that primary and secondary education provisions should be maintained as public goods, and the debate for tertiary education is generating heat [6,7]. Despite a consensus being made in favour of primary and secondary provisions to be public goods, many traditions, practices, and policies have prevailed, suggesting that the public good concept in education, both for primary and secondary provisions, is merely a theory that remains isolated from practice [8].

Primary education, known as universal education, covers some common subjects that all students should learn [7]. Streaming systems have also been identified as clustering education, an affair that often commences in the secondary provision [8]. Many countries follow a rigid clustering system including science, business, and the arts [8]. Students studying in the clustering system in many developing countries must consider one of the clusters from the three [8,9]. Science is considered a favourable cluster because students from the science cluster can always move to other clusters (for instance, to business or the arts) in order to pursue further studies [8,9]. Moreover, graduates from the science cluster can be employed for any job regardless of area of expertise [8,9].

On the other hand, students from the business and arts clusters should only be able to pursue further studies in their own cluster, and subsequently, they can only be employed in one specific area [8,9]. Therefore, secondary science education in many developing countries is a "hot-cake" compared to its business and arts counterparts that may bring fortune because, as a doctor, scientist, or engineer, an individual can also occupy the highest position of public administration that serves the interest of public policy or philosophy [8,9]. Studies have been conducted and have found that this unsustainable and unsubstantial development of education has created a greater level of both vertical and horizontal mismatch, forcing wide misuse of public funds [8–10]. However, no significant steps have been taken to resolve this issue, especially in developing nations. Hence, one largely accepted proposition may suggest that, because elites are mainly studying science programs in emerging nations, this hinders the sustainable development of education, especially science education in the secondary provision [11]. This uneven development does not support all students equally regardless of their SES [12,13].

Science education requires a higher budget compared to its business and arts counterparts [11]. Therefore, misuse and mismanagement in science education are more endangering compared to its other counterparts, though no misuse and mismanagement in education should be tolerated by any means [11,12]. Within all types of education, science education is considered the most powerful tool for sustainable development, as it accelerates both the national economy and individuals' prosperity [11]. Hence, if students with privileged SESs are able to succeed by studying science, and if students with lower SESs are deprived in the name of tradition, practice, or policy, this hampers the sustainable development of education in the era of SDGs, which is the core problem of this study [11–13]. After identifying the research problem, the existing gap in the literature is mapped so that the scope of this paper can be explored before determining the research objective and questions.

*Research Gap and Scope: Objective and Questions*

STEM (science, technology, engineering, and mathematics) is an important field and has been widely studied in the last three decades [14], and it is a broader term used to group several disciplines. Hence, the term is more applicable to tertiary education [14]. The science cluster of the secondary provision in many countries also represents STEM [14]. Thus, the term secondary science education is used in this paper. Studies in connection to STEM were initiated in the mid-1900s [14]. Since then, several movements have been made to transform attention and focus [14].

Most studies conducted since the mid-1900s to early 2000s in STEM for the secondary provision were devoted to discovering substantial courses and curricula so that competent graduates are produced who are able to undertake the challenges involved in higher education [14,15]. The pedagogical affairs and instructional technology of STEM received much attention starting in the early 2000s. Soon, the STEM community also realised that a significant portion of students had dropped from the STEM provision [16]. Thus, some studies have recently focused on how to handle this drop-out issue [16]. According to [14–16], a few studies have also focused on discovering a mechanism of how to make these drop-out students ready for the work of world (especially semi-skilled provision). Lately, gender perspectives and geographical distributions have been considered within STEM studies [16]. However, studies should also be conducted to examine the SES of students studying the science cluster [14–16]; this comparative study is the first attempt to fill this gap, especially in developing countries of southern Asia, such as Bangladesh.

Having identified the research gap and scope, the aim, objectives, and research questions are outlined here. This research aims to discover the SES of the students who are studying secondary science education by comparing three groups, namely science, business, and the arts. Three objectives are presented to achieve the aim, as follows: to explore the SES statuses of students studying in the secondary provision to determine the statuses of students in the science cluster; to analyse the academic performance of students studying in the secondary provision in order to identify the performance of science cluster students; and to establish evidence for whether SES background influences access to and performance in secondary science education programs. By accomplishing these objectives, the following research questions can be answered with regard to the secondary provision:

RQ 1. Does SES matter regarding access to science education?
RQ 2. Does SES impact performance in science education?
RQ 3. What is the impact of SES on science education?
RQ 4. How can sustainable education be developed in the era of SDGs?

After the introductory section, a literature review is presented before the accuracy of the adopted research design is justified. As the paper progresses further, findings and discussions are reported before implication and concluding remarks of this study are noted.

## 2. Literature Review

This section firstly critically explains the concept of development before the relationship between development and education is identified. The literature review concludes by discussing the effect of developmental goals in education, especially from the perspectives of the MDGs (Millennium Development Goals) and SGDs.

### 2.1. Economic and Social Development—Human Needs Perspectives: Capitalist vs. Socialist

Discussion on national development commenced in the early 1950s. Since then, three schools of thoughts have been established, namely economic development, social development, and human needs perspectives [6,7]. Economic development refers to the income capacity of a particular country, which is measured by a number of indicators, such as GDP, GNP, and GNI [6,7]. Economists have heavily emphasised economic development in ensuring national prosperity [16–19]. Owing respect towards the growing popularity of this idea, most countries around the globe have started jumping into global economic competition since the late 1960s to achieve the targeted numbers for economic indicators in their respective policy documents [20].

To achieve the set targets/goals for economic development, some countries have heavily relied on innovation, technology, and capital-intensive goods [6]. Other countries have depended on the export of natural resources (to the countries mentioned prior), and others have had no choice but to rely on the export of labour-intensive goods or "body drain" to the above-mentioned countries [6,20]. The race of economic indicators might have provided both modernisation and commodification by producing a number game for developmental discourse, thereby developing a capitalist ideology [6]; however, it

has certainly failed to reduce the economic disparity amongst different nations, instead substantially widening the gap [6,19].

Ideally, such a global economic race should be considered global politics, exploiting natural and human resources in the interest of leading nations that are technologically and innovatively advanced, as argued by [6,19,20]. On the other hand, this race might have helped a small number of people from the other two country groups to be extra-ordinarily affluent overnight, which subsequently has also supported the growth of their national economic indicators at the cost of huge internal economic disparity within their countries [6]. Refs. [6,19] further argued that social development is an important prerequisite for economic development to be sustainably functional. Moreover, [21,22] noted that, without ensuring social development, the benefit of economic development is no longer effective.

Since the early 1980s, scholars have started heavily arguing in favour of social development [22–24]. A well-known academic, Mahabub Ul Haq, developed a mechanism known as the HDI (human development index) under the supervision of the United Nation to measure national development [6,23]. The HDI includes a number of indicators in addition to economic indicator(s) [6,23]. As claimed by [6,23,24], these additional indicators (such as health, political transparency, education, freedom of speech, etc.) represent the social development status of a particular country.

The measurement of national development via the HDI started receiving popularity in the 1990s, as it was highly advocated by the United Nation [6]. Since then, most countries have transformed their development plans and strategies in order to respond the achievement of the HDI [6]. The robust response towards the HDI made by UN signatories helped them to achieve their desired targets via a "number game" [6,18,23]. However, whether such achievement of targets via the "number game" ensures human needs perspectives is an unsettled debate [25]. Hence, human needs perspectives began as a new item in the concept of national development in the mid-1990s [26].

The concept of national development, its criteria, and its schemata was designed in the West [6,26]. International developmental thoughts developed in the West are implemented in Western countries by their respective governments [6]. The government of a Western country might choose one path/approach, namely the "capitalist" approach, the "socialist" approach, or a blend of the two, in order to respond to national development strategies [27,28]. However, regardless of which approach is considered in a Western country, human needs perspectives remain a central focus [28]. Therefore, the development strategy of a Western country might follow the "capitalist" path for an international audience, and a local audience may benefit from some policies that support "socialist" views [28,29].

Although the developmental strategies of Western countries are executed by their governments, countries in the East depend on the supervisory role played by development partners, especially the United Nations [30]. Hence, countries in the East lack the required competency and voice to explore a substantial development strategy and approach, forcing them to be reliant on the UN and developmental partners (World Bank, ADP, USAID, etc.). The UN and development partners are the "foot soldiers" of the West [31], precisely for sponsoring countries. Consequently, the developmental strategies and approaches that are considered in an Eastern country under the supervision of the UN are not distinct in nature for pursuing the substantial development of a particular country/region [32,33].

The UN often offers a "common prescription" for its signatories' developmental strategies and approaches (labelled as developing/emerging) [33]. Currently, the UN's prescription has moved from the MDGs to the SDGs [32,33]. Whether a common UN prescription helps signatories (labelled as developing/emerging) to achieve sustainable development is hard to determine [31–33]. Indeed, the conjointed development agenda of the UN has forced its signatories, known as developing countries (followers), to follow the dependency theory [28,34]. Here, to conclude the literature review, the impact of the MDGs and SDGs on the development of UN signatories is explored with a particular focus on education. The following examines the relationship between development and education.

### 2.2. Relationship between Development and Education

Studies conducted in the area of development and education primarily investigate the role of education on development [6]. Although almost all evidence supports that education greatly supports national development, very few studies [6,28,35] have claimed that education aggravates the adverse effect of the dependency theory in the name of enhancing national development. However, education is considered the most powerful tool for national development [6,35]. Is education itself an item for development agenda, or is it a tool for development? This is a discourse that has never been brought to light [6,35].

The previous discussion (Section 2.1) illuminates that development is an idealist concept [28,29]. This idealist concept has become an intangible good, which has also become an export item [32] that is diplomatically freighted to countries that are labelled as followers [32,33]. Under such a climate, it is important to study the relationship between development and education. However, this topic lacks attention because the role of education on national development experiences dominating focus [6,28]. The following does not communicate evidence from field research; rather, it is a critical discussion using the literature to logically map the relationship between development and education.

Without generating further debate, it is humbly acknowledged that education always supports both economic and social development, as well as human needs perspectives. Either a comparison between educated and non-educated counterparts or an evaluation of the conditions of pre- and post-education eras would surely assert that education supports the development of national and individual levels under any circumstance [6,28]. However, research covering investigations of the competitive advantages received from education by leading nations (developed country) and following nations (developing country) is yet to be popularised [6,28]. It is thus likely an important time to conduct such research, an appeal that is spotlighted by the following discussion.

Firstly, education, especially the tertiary and secondary provisions, often help learners accrue necessary skills and competencies that are needed for the work of world, as argued by [36,37], but this is, in fact, not factual for every situation [6]. However, even if the claim made by Bogviz et al. 2020 [36] and Buchanan et al. 2015 [37] is 100% factual, the skills and competencies delivered by Eastern institutes are borrowed from their Western counterparts, aggravating the adverse effect of the dependency theory [6,28]. Education in Eastern countries serves as an agent, not as an innovator or manufacturer [6,28]. Regarding education that is not capable of producing leaders and innovators but that may develop agents, would such a particular education be able to offset the adverse effect of the dependency theory? This is a question that is yet to be answered [6,28,32].

Secondly, education, in particular referencing the primary and secondary provisions, provides necessary attributes, skills, and competencies to pupils in order to boost social development [6]. Are these attributes, skills, and competencies related to social development the fundamental parts of theoretical learning? Or is this a practice that is driven by society and context? This issue often generates heated discourse [6,28]. Adopting attributes related to productive cultures, decent traditions, equality, equity, inequality, social justice, good governance, and political ideology cannot be injected using only a borrowed theory from the West [38]. Hence, education remains within the agenda of social development, where society and context need to shape education for self-motivated demands, not only using education to obtain degrees for personal gain [5]. Theory-driven learning on social development, which is isolated from contextual social norms and practices, may produce a small elite society without making many contributions to influence positive change in social development [23].

Finally, both public and private education consume formal and informal budgets [39]. Investment in education is large for a country, and it is often found that investment in education is significantly worthwhile [38]. Although it is urged that more investment in education is badly needed, it is further argued that the finance of imported education that develops dependency is utterly a bad investment [32,33]. Hence, it is also suggested that contextual innovation is important for the development of education in order to mitigate

the specific problem that a country faces without being heavily reliant on a common prescription made by the UN [6,40]. One prescription may help leader countries to export international development as an intangible product, where the education of emerging nations can be an agent without being an innovator or manufacturer [6]. The following explores the effect of international developmental goals (especially MDGs and SDGs) on the education systems of developing countries.

### 2.3. Developmental Goals: MDGs to SDGs—Science Education and Elitism in Education

Before introducing the MDGs in 2000 and SDGs in 2015, ratifications and agreements (such as EFA, GATS, UNFCC, etc.) were the prime tools used by the UN to monitor and supervise its signatories [30,31]. After establishing the MDGs and SDGs, both the terms and references set for developmental goals and existing ratifications, as well as agreements, were used to supervise UN signatories (30, 31). Moreover, new ratifications and agreements are produced from time to time in order to ensure the achievement of developmental goals [40]. The following does not necessarily explain the entire phenomena of the developmental goals of the UN; rather, it exclusively focuses on education.

Considering education as a public good, a public exchequer can be used to cover it in order to avoid commercialisation and commodification in education [6]. Following the GATS' agreement prescribed by the WTO, many developing nations introduced a private education sector in the early 1990s. According to the GATS verdict, the signatories should remove the trade barrier from their education systems, allowing commercialisation [33,34]. When the MDGs commenced in 2000, EFA (education for all) was a UN fundamental mandate for education [6]. According to this mandate, all signatories must provide universal education to every child regardless of their SES, race, ethnicity, and gender [4–6]. To achieve this target, signatories are advocated to expand their education sectors dramatically via NGOs and private provisions [5,6].

This approach was upheld by arguing that this would subside the burden placed upon governments [6]. Although students from privileged SESs would obviously go to the private sector, poor pupils could receive education both from NGOs and their public counterparts [41]. This strategy was further prescribed, suggesting that public anarchy in education must be driven out [6,7]. With the support of this approach, multiple providers were welcomed to compete, and education became a "commercial commodity" [41]. The concept of "unified education" for national interest was challenged in many ways [41,42]. The strategy was at least largely successful in allowing access to universal education for everyone. However, this strategy was also silently ensuring freedom of choice in education for elites only via a response to the capitalist view [42].

Enrolment in education was dramatically increased because of a collective force working to ensure EFA [41,42]. This sudden increased enrolment also created enormous pressure for secondary and tertiary provisions (6, 7). The secondary provision, especially higher secondary and tertiary education counterparts, were overwhelmed [12,13]. To resolve this crisis, the governments of UN signatories were advocated to expand their private education sectors drastically, especially for the tertiary and secondary provisions [15]. Moreover, informal investment (such as private tutors, support from private coaching centres, and additional support from teachers in exchange for private funds) was encouraged particularly to learn science education because, being an international product, science education is more expansive compared to its arts and business counterparts [14].

Financially solvent parents were happy with this approach, making science education a luxury product that only they could buy to ensure a better future for their children [16]. Poor children had no choice but to be reliant on public funds, which might be insufficient for any form of education, let alone science education [6]. Ref. [16] further noted that, as a consequential effect of the MDGs (especially EFA), a commercially viable private higher education (HE) sector was established that also follows a rigid capitalist view to ensure education for the elite [12]. Elitism in education is not a recent phenomenon [1]; however,

the above discussion suggests that the MDGs era has brought a new horizon for the concept of elitism in education.

As claimed by the EFA monitoring team, the MDGs era was largely successful in meeting the EFA targets [1]. The team has, however, realised that the drastic and rapid expansion in education as a result of MDG policies has primarily ensured quantitative success [1]. Therefore, the team has advocated in favour of quality education as part of the strategy outlined in the SDGs [6]. To ensure quality education, a number of agendas (such as objective-based learning, OBE for HE, TVET education for the poor, technology-based learning and assessment, blended learning, and sustainability technology in education) have been outlined [6,16].

Whether the education agenda of the SDGs really improves education quality or promotes a capitalist view in emerging countries' education is a matter that deserves more research attention in the era of the SDGs [6,28]. For instance, the purpose of HE is to produce leaders, innovators, and critical thinkers [12]. Would massive and monopolistic adoption of OBE meet the authentic purpose of higher education, or would it further aggravate the adverse effect of the dependency theory in education? An answer to this question is badly needed before responding to the SDGs. HE and research not only shape the entire education system of a country but also help it progress further [6]. Adaptation to the dependency theory in education must welcome more adverse challenges, which would certainly hinder sustainable development in education [28].

### 2.4. Convention against Discrimination in Education 1960 and Interventions: Flashback

In an era of the ongoing popularisation of education, scholars have asserted that education not only supports national development and public welfare but also ensures private benefit [6]. Thus, discrimination in education in order to obtain private gains has become a more serious concern since the early 1940s [11,12], which was the main enemy for the substantial function of education. To address this crisis, UNESCO imposed a ratification labelled the Convention against Discrimination in Education 1960 for its signatories. According to this convention, any form of discrimination in education cannot be tolerated under any circumstance. Such a convention obviously helped various minority and vulnerable groups access education (based on race, ethnicity, gender, and physical and mental condition), which has created and extended the market of education [1].

Additionally, a number of global interventions, such as EFA, the MDGs, and the SDGs, have also helped to extend the market of education [1,3,6]. Using the scope of such an extended market, a market-drive theory has been slowly built by replacing philosophical thoughts in education [5,6]. Hence, education has become a commercial product/commodity instead of a program that is operated based on philosophy and principle. Today, the market determines every phenomenon in education [7,8]. This attitude has not only developed various types and patterns of education but has also created education with a different quality and purpose [3,4].

Several international interventions in education (such as EFA, the SDGs, and the MDGs) needed to be customised for contextual fitting. Hence, non-customised interventions have made education in some emerging nation a "class-based" program instead of a right-based agenda in which the capacity and interests of students are not the prime parameters, rather regulating student choices in education according to SES [7,8]. This has silently developed an anarchy in education which reminds us to remember the UNESCO Convention against Discrimination in Education 1960.

## 3. Research Context

The following explains some relevant information on the research context, namely Bangladesh. Firstly, the secondary education system, which helps to justify the research focus, is described. Secondly, the effect of the MDGs on the school system in Bangladesh is noted. This information also supports the justification of the accuracy of the adopted research design, as explained in the next section.

### 3.1. Secondary Education

Four types of providers deliver secondary education in Bangladesh, of which one is known as the general secondary provision, which covers almost 81% of students [8–10]. Another provision known as madrasa education is the second largest counterpart, which covers almost 12% [10]. The remaining two types of education, namely vocational education and international education, cover 5% and 2%, respectively [42]. These two providers are completely isolated from the general system [43]. Only the super elite group, which is very small in number, pursues international education, such as American and British models of schools operated in Bangladesh. The extremely poor group either goes to the madrasa or vocational provisions [42]. Hence, general secondary education is a major part of the country's secondary education, which is the subject of this study, as explained below.

General secondary education, which is divided into three steps, is a seven-year program that starts in grade six and ends at grade twelve. Grades six to eight are labelled as the junior secondary provision, whereas grades nine to ten and eleven to twelve are labelled as the secondary and higher secondary provisions, respectively [44,45]. Universal education, also known as compulsory education, has recently extended to grade eight from grade five [42]. Vocational education starts in grade nine. After completing grade eight, students may shift to the vocational path if they do not want to continue with general secondary education [46].

Students who move to the vocational system are not allowed to re-move to the general system. However, students in the general system who study science programs are allowed to move to the vocational path or any desired path at any point in their education career [46]. Students need to attend a public examination known as the secondary school certificate (SSC) at the end of grade ten [8]. Students who successfully complete this examination achieve the status of an SSC (general) graduate. Successful students from the vocational path achieve the status of an SSC (VOC) graduate who can only pursue the polytechnic provision to continue further studies [46].

SSC (general) graduates from the science path either go to the higher secondary provision or the polytechnic provision to pursue further education [45]. Furthermore, SSC (general) graduates from the science path can study science, business, or the arts in the higher secondary provision, and their counterparts, i.e., students from the arts and business paths, must remain on the same path if they continue to study higher secondary education [8]. Selecting a cluster or path starts in year nine in Bangladesh [8–10]. Students studying a particular cluster must study the subjects that are determined for his/her chosen cluster [8–10]. The system is not liberal enough to allow studying various subjects covering multiple areas, such as science, business, and the arts [8].

Although students in the secondary provision in many developed countries enjoy the liberty of choosing their desired subjects from different areas (i.e., science, business, and the arts), students in some developing countries, including Bangladesh, are restricted to consider one cluster [8–10]. With policy support, students in the science cluster can pursue higher education (HE) in any area, whereas students from other clusters are restricted to study higher education in their respective area only. The scope of HE and the job market in developing countries is very narrow [9]. The limited scope of HE and jobs in the area of business and the arts is occupied by science graduates as a result of one-way traffic that favours the science cluster [10]. This has made the science path a "hot-cake" in the secondary provision, and students from higher SESs study science to achieve the SSC at any cost to deprive others, which is the subject of this study [8,10].

### 3.2. The Effect of MDGs and SDGs on Secondary School

Before the introduction of the MDGs, particularly the EFA agenda, both public and semi-public schools delivered secondary education [42,43]. The government exchequer provided 100% and almost 90% of the budget of public and semi-public schools, respectively [12]. After the introduction of the MDGs, both NGO and private school provisions were added as providers of secondary education [42]. Private schools, which are funded

by the students' parents, are in the elite/affluent areas of an urban geography [42]. NGO schools funded by development partners are in poor areas of both urban and rural geographies [41,42]. This approach was considered to ensure extensive coverage so that students from all economic backgrounds can be covered [42]. Although NGO schools cover students from lower SESs, their private counterparts provide access to students with higher SESs [42,43].

Ideally, public and semi-public schools would provide access to students regardless of their SES. Before the MDGs, the expansion of public and semi-public schools was based on political interference only [43]. Activities related to the MDGs have managed to minimise political interference for the establishment and development of schools [41]. The MDG team working both in the UN and in the Bangladesh government system managed to develop some models for the establishment/development of schools and their funding mechanisms [43].

These models concentrated on regional and demographic affairs from various angles to establish new schools and to develop older schools so that both high-income and middle-income groups, as well as low-income groups, can access education in a particular public or semi-public school [45]. One of the main setbacks of these models is that a public/semi-public school is no longer a common place for students from all kinds of economic backgrounds; rather, a particular school is dedicated to a specific group of people [46]. On the other hand, one of the outstanding advantages is that these models offer help to determine whether students from lower SESs are attending schools, and based on the information gathered, subsequent follow-up actions can be taken to ensure access for students from lower SESs [12,46].

The MDG team further developed a few models for data collection and analysis so that the real situation of access to education and performance in education made by different groups (such as those depending on economic factors, gender, and location) of students can be traced easily [8,41]. The data management system in education, especially for primary and secondary provisions, was substantially improved by the MDGs [47,48]. Having said that, the MDGs should not be blamed for the silent development of the elitism of science education in Bangladesh, as it was not within the purview of the MDGs. The policymakers of the country should have taken measures to address specific problems that Bangladesh faces, which is a statement given by a member of the MDG team [47].

The effect of the SGDs is yet to be explored because the groundwork of the SGDs only started in late 2016. Soon after their commencement, COVID-19 affected the education system in 2020 and continues to do so. However, most elite schools started adopting technology-based learning and assessment, blended learning, and sustainability technology in education following a dependency model [28,36,37]. The University Grants Commission (UGC) of Bangladesh issued a verdict, in which all universities must follow OBE without paying due attention to the effect of OBE on higher education [12].

## 4. Research Design

Firstly, the reason for adopting a qualitative method is justified before the involved data collection process is explained. Prior to explaining the primary data in the final two sub-sections, the development of the used domain and data analysis for secondary sources is described.

### 4.1. Methodological Approach: Justification for a Qualitative Method

Given the nature of the outlined research questions, a qualitative method is used, adopting data collected from both secondary and primary sources. Before explaining the data, the reason for using a qualitative method is first justified.

Of the four research questions outlined in the introduction, none are able to be used to develop a hypothesis that can be considered either null or alternative. Hypotheses are often tested to understand the norms of relationships that may be either positive, negative, or casual [49]. To test these relationships, different quantitative techniques are

often used [50]. Thus, a quantitative method cannot be adopted [51,52]. For instance, if this study considered a hypothesis regarding whether the rich are only studying science, a quantitative technique would be able to establish a relationship.

Instead of testing such a hypothesis, this study aims to discover insight into what kind of SES background that students have who are studying different clusters (such as science, business, and the arts) in the secondary provision. The aim of this comparative discourse (comparison made among three clusters) is to generate possible directions that future studies can use to develop hypotheses—a core purpose of a qualitative study [53]. Hence, this qualitative study adopts a descriptive analysis of data collected from both secondary and primary sources. Why and how the descriptive analysis was made are explained later.

*4.2. Area and School Selection for Secondary Data*

Currently, Bangladesh has eight divisional cities, namely Barisal, Chittagong, Dhaka, Khulna, Mymensingh, Rajshahi, Sylhet, and Rangpur. Bangladesh commenced its journey just after gaining independence in 1971 with four divisional cities, namely Chittagong, Dhaka, Khulna, and Rajshahi, whereas the others were the largest municipalities except Mymensingh. These largest municipalities were gradually added as divisional cities. Mymensingh was the latest city added as a division in 2015. Because the impact of the MGDs and SGDs is still being witnessed, Mymensingh is not included in the sample area. All divisional cites are considered because other cities within a division ideally follow its divisional counterpart in the trend of development.

All divisional cities have urban, semi-urban, and rural areas [42]. Following suggestions from the MDG agenda, schools are classified as upper class, middle class, and lower class, located across the country. For example, all kinds of school are situated in the urban, semi-urban, and rural areas [41,42]. However, urban areas should ideally have more upper-class schools compared to their rural counterpart, and a significant number of lower-class schools should also be situated in urban areas [42]. For example, Dhaka is the capital city, and it is also known as the most urbanised city in Bangladesh, as it is dominated by lower-class schools and many poor people in urban slums, settlements, and tin sheds [42]. Hence, all schools that were established by 2010 in these divisional cities are considered.

The data on the appearance and performance on the SSC are the key subject of this research. Students who were admitted to grade nine in 2010 should have taken the SSC examination in 2012. Hence, the SSC data of all schools located within the seven divisional cities since 2012 were used. Data from every four years was used, i.e., data from 2012, 2016, and 2020. It was hard to collect data from every year and input them into a computer, which is why data from every four years were used.

A total of 3315 schools are involved in the seven divisional cities, of which 887 schools are in urban areas, and the others (2438) are located in rural areas (Figure 1). Of 3315 schools, the number of upper, middle, and lower-class schools are 138, 833, and 2344, respectively (Figure 1). A total of 925,717 students participated in the SSC from 3315 schools (Figure 1), of which upper, middle, and lower-class schools had 120,577, 389,254, and 415,786 students, respectively (Table 1). Of 925,717 students, years 2012, 2016, and 2020 had 252,295, 303,340, and 369,982 students, respectively (Table 1). More information on the sharing of schools and students by the seven divisional cities can be found in Figure 1.

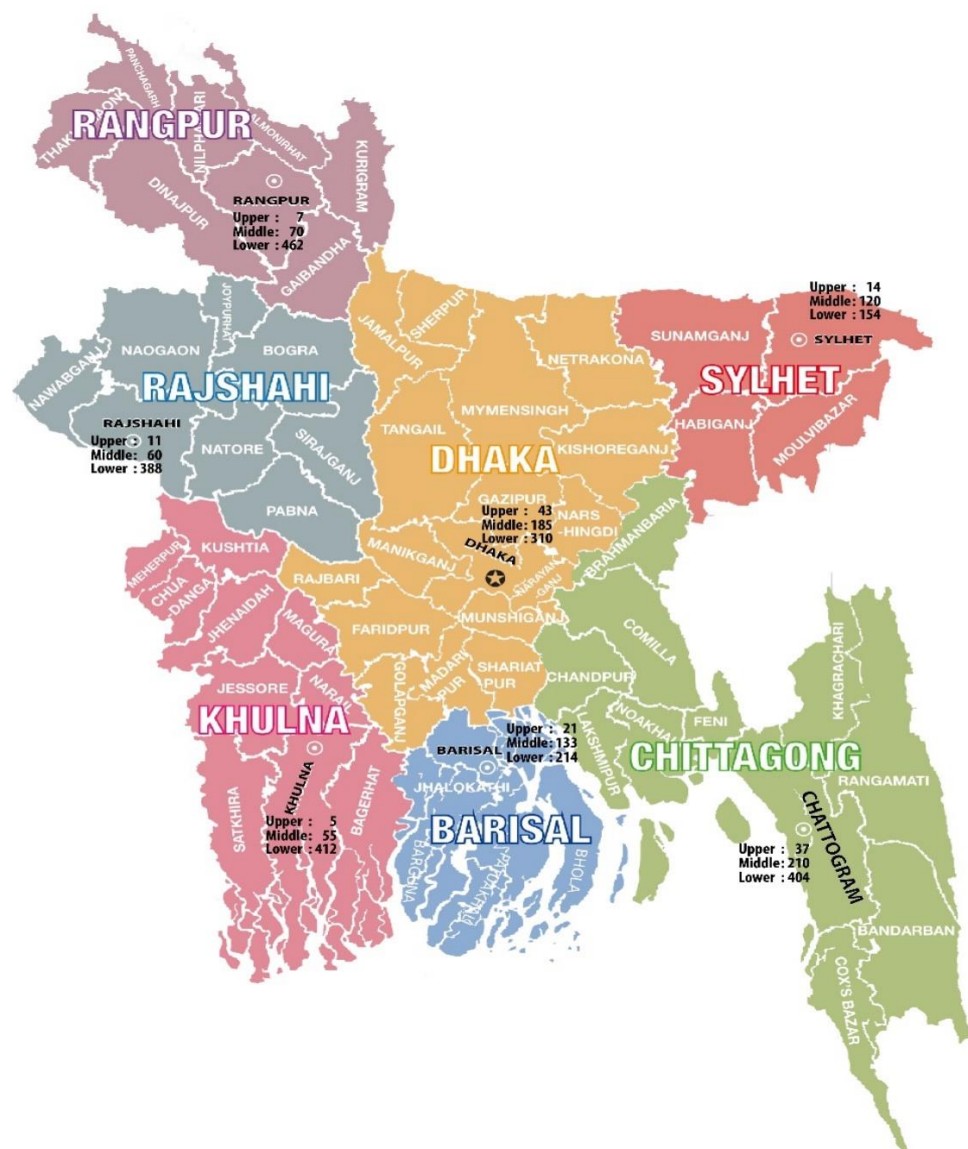

**Figure 1.** *Number of schools by location and their status.* Black-coloured fonts represent location and status-wise numbers of schools in different districts.

**Table 1.** Number of students by year and economic class.

| Year | SES | Total Appear | Total Pass | Business | | Arts | | Science | | GPA 5 | | |
|------|-----|-------------|-----------|----------|------|------|------|---------|------|----------|------|---------|
| | | | | Appear | Pass | Appear | Pass | Appear | Pass | Business | Arts | Science |
| 2020 | Upper class | 48,378 | 45,816 | 11,968 | 11,014 | 5543 | 4398 | 30,867 | 30,404 | 674 | 141 | 17,645 |
| | Middle class | 167,746 | 144,872 | 56,003 | 49,057 | 61,730 | 47,909 | 50,013 | 47,906 | 757 | 248 | 13,984 |
| | Lower class | 153,858 | 130,506 | 42,464 | 36,986 | 73,430 | 58,150 | 37,964 | 35,370 | 390 | 281 | 5697 |
| 2016 | Upper class | 39,711 | 38,451 | 11,679 | 11,000 | 3045 | 2651 | 24,987 | 24,800 | 908 | 77 | 16,087 |
| | Middle class | 124,322 | 112,871 | 52,183 | 47,643 | 34,477 | 28,872 | 37,662 | 36,356 | 1064 | 94 | 10,721 |
| | Lower class | 139,407 | 123,978 | 49,507 | 44,740 | 56,354 | 47,458 | 33,446 | 31,780 | 611 | 95 | 5114 |
| 2012 | Upper class | 32,488 | 31,203 | 12,039 | 11,336 | 3,112 | 2748 | 17,337 | 17,119 | 2122 | 140 | 11,468 |
| | Middle class | 97,186 | 86,237 | 45,205 | 39,842 | 27,875 | 23,521 | 24,106 | 22,874 | 2179 | 235 | 7085 |
| | Lower class | 122,621 | 105,090 | 48,562 | 42,004 | 51,051 | 41,806 | 23,008 | 21,280 | 1024 | 204 | 2821 |

**Table 1.** *Cont.*

| Year | SES | Total Appear | Total Pass | Business | | Arts | | Science | | GPA 5 | | |
|---|---|---|---|---|---|---|---|---|---|---|---|---|
| | | | | Appear | Pass | Appear | Pass | Appear | Pass | Business | Arts | Science |
| | Upper class | 120,577 | 115,470 | 35,686 | 33,350 | 11,700 | 9797 | 73,191 | 72,323 | 3704 | 358 | 45,200 |
| Overall | Middle class | 389,254 | 343,980 | 153,391 | 136,542 | 124,082 | 100,302 | 111,781 | 107,136 | 4000 | 577 | 31,790 |
| | Lower class | 415,886 | 359,574 | 140,533 | 123,730 | 180,835 | 147,414 | 94,418 | 88,430 | 2025 | 580 | 13,632 |
| | **Total** | **925,717** | **819,024** | **329,610** | **293,622** | **316,617** | **257,513** | **279,390** | **267,88** | **9729** | **1515** | **90,622** |

Given the individualistic nature of the research question, multiple types of data with different natures and differentiated tools and analyses were used to respond to the need of each research question (Table 2). Hence, RQ1 and RQ2 are answered via secondary data, whereas RQ3 and RQ4 are answered via further discussion of RQ1 and RQ2 and via primary data (Table 2). The following discussions note the development of domains for secondary data, their sequential collection process, and data analysis.

**Table 2.** Data collection tools for individual RQ.

| RQ | Primary Tool(s) | Auxiliary Tool(s) | Method |
|---|---|---|---|
| Does SES matter regarding access to science education? | Domain and tool development; data collected from BANBEIS and WBRSEB archive based on the developed domains | Percentage analysis according to different domains | Qualitative (Descriptive analysis of secondary data) |
| Does SES impact performance in science education? | Domain and tool development; data collected from BANBEIS and WBRSEB archive based on the developed domains | Percentage analysis according to different domains | Qualitative (Descriptive analysis of secondary data) |
| What is the impact of SES on science education? | FGDs with different stakeholders who are the subjects of the domain | Literature review, interpretation of the findings, and discussion of earlier research questions | Qualitative (Discourse and narrative analysis) |
| How can sustainable education be developed in the era of SDGs? | FGDs with different stakeholders who are the subjects of the domain | Literature review, interpretation of the findings, and discussion of earlier research questions | Qualitative (Discourse and narrative analysis) |

*4.3. Domain (Instrument) Development: Secondary Data Collection, Analysis, and Reliability*

A pre-determined domain, known as school classification (upper, middle, and lower), used for monitoring the agendas of the MDGs and SDGs, is the key subject and fixed domain of this research. The second domain, known as clustering (science, business, and the arts), is used to understand the distribution of the students within the different categories of the first domain. This can help to understand whether a particular class of students (upper, middle, and lower) dominates within the second domain, known as clustering (science, business, and the arts). This helps to understand the influence of SES on education choice, a prime inquiry (RQ1).

The third and final domain is the academic result. The academic performance domain, which is divided into three categories, namely fail, pass, and grade point average (GPA) (GPA 5 is considered the best performance), is checked against the other two domains,

namely school classification (upper, middle, and lower) and clustering (science, business, and the arts) in order to understand whether academic performance is controlled by the other two domains. This would certainly provide information that can be used to examine the role of SESs on program choice and the subsequent effect on academic performance, an inquiry made in RQ2 (Table 2). Now, the processes involved in collecting the data are presented.

BANBEIS (Bangladesh Bureau of Educational Information and Statistics) is a UNESCO patronised public institute and is responsible for the management of data related to educational policy and planning. Fundamental information on Bangladesh education is accessible via the BANBEIS website. In-depth data related to a particular study can be collected via contacting the library wing of BANBEIS. A classified list of schools (upper, middle, and lower-class schools) located in the seven divisional districts was collected from the library wing.

A public archive named Web-based Results Publication System for Examination Boards (WBRSEB) preserves the information regarding the SSC and HSC examinations of students from each school in Bangladesh. Firstly, a list of students and their SSC registration numbers (SSC ID No.) was collected for each school for a total of 3315 samples (covering upper, middle, and lower classes) via the website for the years of 2012, 2016, and 2020. This provided us the data of students studying in different types of schools (upper, middle, and lower classes) with their geographical locations (such as urban and rural).

Using the SSC ID, each student's chosen cluster (science, business, and the arts) and their achieved academic performance were traced. This is how information was obtained on the chosen clusters (science, business, and the arts) and academic performance of each student from every school. Subsequently, information of all schools regardless of their status (upper, middle, and lower classes) was obtained. This information can be further segregated (divisional diversity, economic diversity, and location—urban and rural) in different ways as required for data analysis. However, school classification (upper, middle, and lower classes) remained as the focus of data analysis.

This research neither tested null and alternative hypotheses nor developed a causal relationship between different variables, as presenting mathematical or statistical models (such regression or SEM—Structural Equation Model) is not important [51–54]. Moreover, selective samples were not used, nor were the mean, median, and average values obtained for every group. Rather, information was collected from each student from all available students in a particular year. Moreover, no testing and comparisons were made among different years either to develop a causal relation or to assess a hypothesis. However, no intended comparison was made to test a particular issue (such as equity, inequality, and diversity amid different demography) among the seven divisions. Hence, as argued by [55–57], under such a climate, various statistical tests (such as the "*t*-test", "z-test", "chi-squared test", "ANOVA test", "binomial test", "one sample median test", etc.) were no longer relevant.

Information from each respondent was firstly deposited independently, which then led to a total value. This independent information was not based on a perception, but rather on a fact that resulted in a number. The numbers were generated based on real fact thats finally contributed to a total. Subsequently, the totals were distributed to the domains according to a percentage of the total. Under such circumstances, as argued by Bolster [56], no "significance test" that is used as a statistical parameter is required. An analysis via presenting a percentage is one of the most acceptable ways for this kind of analysis, as argued by [56,57]. This approach should be labelled as "descriptive analysis"—an important tool to develop further hypothesis for future research [56,57].

### 4.4. Sampling and Triangulation for FGD: Primary Data

Primary data were collected via focused group discussions (FGDs) in the second phase. The first phase collected the secondary data and analysed them before the second phase started, which is considered an appropriate approach, as argued by [51,52]. The

purpose of these primary data was not to quantify any perspective or perception [51,52]. RQ3 and RQ4 concentrate on having some deeper understanding of the affairs exposed by RQ1 and RQ2. Hence, any form of quantification is not a focus; rather, "fact finding" for deeper understanding was the main objective [53,55]. Consequently, FGDs were considered to provide in-depth information [54,55]. Because quantification was not a concern, the number was not a prime pre-condition for sampling [55]; rather, "triangulation" through representative samples was the matter of emphasis [54].

A total of 27 FGDs were conducted with three groups (students, parents, and teachers) involved in three clusters (such as science, business, and the arts) of three types of schools (upper, middle, and lower classes). A separate FGD was conducted for each cluster of students (i.e., science, business, and the arts) from three types of schools (i.e., upper, middle, and lower classes), resulting in 9 FGDs with students. The same was conducted with parents and teachers, as a result of which 27 FGDs (Table 3) occurred to value the concept of triangulation [51,52,54].

**Table 3.** Sample and sampling.

| Type of Economic Class | Type of Cluster | Type of Stakeholders for FGD | | | Total FGDs |
|---|---|---|---|---|---|
| | | **Teacher** | **Parents** | **Student** | |
| Upper class | Business | 1 | 1 | 1 | 3 |
| | Arts | 1 | 1 | 1 | 3 |
| | Science | 1 | 1 | 1 | 3 |
| Middle class | Business | 1 | 1 | 1 | 3 |
| | Arts | 1 | 1 | 1 | 3 |
| | Science | 1 | 1 | 1 | 3 |
| Lower class | Business | 1 | 1 | 1 | 3 |
| | Arts | 1 | 1 | 1 | 3 |
| | Science | 1 | 1 | 1 | 3 |
| **Total** | | **9** | **9** | **9** | **27** |

### 4.5. Primary Data Collection, Analysis: Confidentiality, Coding and Limitations

FGDs were conducted in the respective schools using the common room, which is often used for the purpose of cocurricular activities. The first round of FGDs was conducted with the students, whereas the second and final rounds were conducted with the parents and teachers, respectively. This sequence was made with the idea that some important facts might arise from the first round that could be clarified with the parents. The final round was conducted with teachers, as they might have been in a better position to illuminate the points that were generated from the first and second rounds.

Before starting the FGDs, the research purpose and focus were explained. The findings of the secondary data were presented in the FGDs so that the respondents could provide us with clarification of the facts from their own experiences and circumstances. A set of semi-structured questionnaires was prepared for each group. However, leading questions were often made to clarify many evolving issues raised within the discussions. The development of a rapport was attempted by providing attention to their responses with eye contract and a positive vibe. However, caution was given in order to not make them feel biased or make them biased. All the FGDs ended with positive conclusions. Before ending, the respondents were asked if anyone wanted to share something that was not discussed but that might be relevant. After this statement, several useful comments were given.

Permission to record the FGDs was sought, and all participants agreed, allowing us to record, transcribe, and listen to the consenting participants' discussions for our analysis. The respondents were assured that the confidentiality of the discussions would be maintained. In the interest of respondent anonymity, codes were assigned for each FGD. Hence, S, P, and T codes were assigned to represent students, parents, and teachers, respectively. SC, BU, and AT codes were assigned to represent the clusters of science, business, and the arts, respectively. U, M, and L codes were assigned to represent schools

from upper, middle, and lower classes, respectively. Hence, for example, an FGD with students from the science cluster from an upper-class school was assigned a full code of SSCU, and the full code for the FGDs with students from middle and lower-class schools was SSCM and SSCL, respectively.

This study also experienced some challenges that are common issues in other FGDs. Some of the respondents who wanted to dominate the conversation were more vocal. This sometimes derailed the conversation and demotivated some introverted respondents to express their viewpoints. To resolve this situation, passion, silence, and close listening were employed. The scheduled time was extended in case anyone wanted talk even after the FGD officially ended. Constraints were faced from time to time under different circumstances, so past experience (three decades in education management) was utilized to the greatest effect possible. Despite all these efforts, some limitations were unavoidable, but they did not greatly impact the novelty of this study. Hence, this study may be a milestone supporting the development of hypotheses for future studies.

## 5. Findings and Discussions

Given the nature of qualitative inquiry, the findings and their discussions are made in parallel before noting the research implications and conclusions in the final section. The first two sub-sections that follow attempts to report on the first two research questions via secondary data, prior to reporting the last two RQs via primary data.

### 5.1. Mapping of Access: Impact of SES on the Choice of Cluster—An Elitist View

Amongst the three clusters (science, business, and the arts), science was the smallest group in 2012, which was 25.5% (Figure 2). Gradually, the share of science increased, as it was 32.1% in 2020 (Figure 2). Although the share of the arts group was quite unstable, the share of the business counterpart constantly declined (Figure 2). According to Figure 2, the share of the business cluster was 41.9% in 2012, which decreased to 29.8% by 2020 (Figure 2). One of the possible reasons for this decrease could be that students wanted to keep both avenues (science and business clusters) open in the HSC provision because only SSC graduates from the science cluster were offered this advantage [8–10]. Ref. [8] found that the share of the business cluster in HSC is the largest, and SSC science graduates shifting to business have contributed to this largest share.

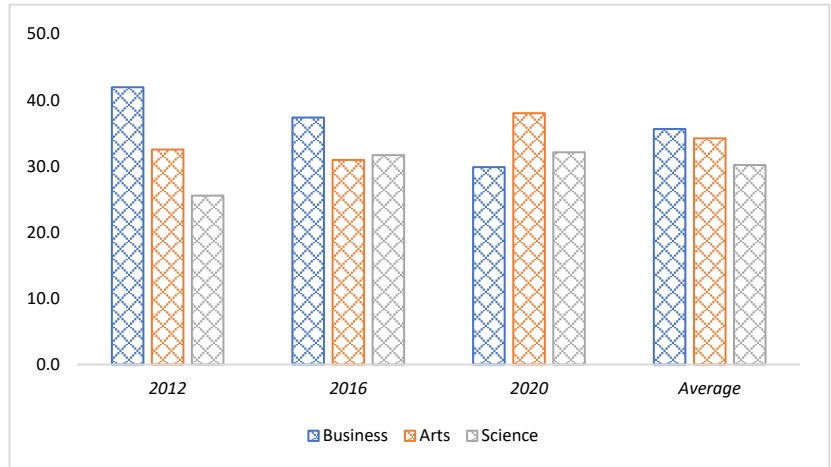

**Figure 2.** *Cluster-wise student sharing in different years.* Horizontal axis represents clusters and years, and vertical axis represents percentage share of students.

However, although science is not the largest group, it has been increasing over time; thus, it is important to understand whether students with any specialised background related to SES contributed to the growth. Before, this issue is discussed further, it should be noted that the number of upper-class schools was lower compared to the lower and

middle-class counterparts. Subsequently, upper-class schools shared a small percentage of student within the total student population (Figure 3). For instance, 12.9% of students were involved in upper-class schools in 2012, which slightly increased to 13.1% by 2020 (Figure 3). A major portion of students studied both at middle and lower-class schools (Figure 3). Both middle- and lower-class schools competed with each other regarding having the highest share of students (Figure 3).

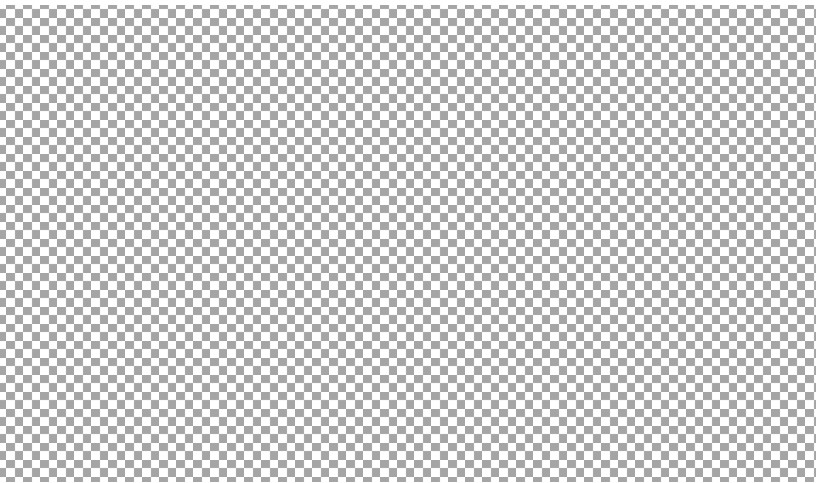

**Figure 3.** *Overall percentage share of students based on different economic classes.* Horizontal axis represents economic status, and vertical axis represents economic class-wise percentage share of students.

Although upper-class schools had a lower percentage of students within the total share, the science cluster was the largest group in this provision. The share of the science cluster has also continuously improved (Figure 4). According to Figure 4, the science cluster of upper-class schools shared 53.4% of students in 2012, which increased to 63.8% by 2020. On the other hand, lower-class schools were mainly dominated by students from the arts cluster, followed by the business counterpart (Figure 4). Middle-class schools were dominated by students from the business cluster, followed by the arts counterpart.

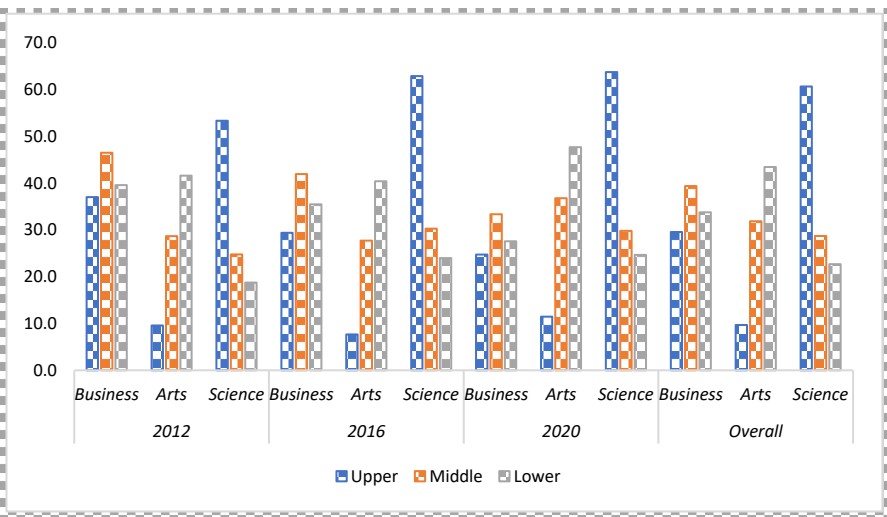

**Figure 4.** *Share of students with different economic backgrounds in three clusters.* Horizontal axis represents cluster and economic background, and vertical axis represents cluster-wise percentage share of students.

Science education not only provides a promising future but also offers multiple avenues for furthering the education and job market, as argued by [8–10]. Therefore, science

education has gained high demand [8–12]. Hence, a shortage of supply to meet the demand made science education as an "elite product" [12,15,16]—a view that is further supported by this study. Should science education be an elite program that follows the economic principle of luxury products? This question deserves greater attention in this era, referring to "rightism in education" or "commodification in education" [1,6,8–10,28]. The above discission highlights how science education has become an elite product in the era of MDGs and SDGs. The following section examines whether an "elitist view in education" is only limited to access or if it has spread with performance.

### 5.2. SES and Education Performance: An Elitist Perspective

Both pass rate and GPA 5 holders are comparatively very high for upper-class schools. According to Figures 5 and 6, the average pass rate for all years (2012, 2016, and 2020) involved by combining every cluster (science, business, and the arts) is higher for upper-class schools, which is 95.76% (Figures 5 and 6), whereas the combined average is 88.47% (Figure 5). The pass rates for upper-class schools in the years of 2012, 2016, and 2020 are 96.04%, 96.86%, and 94.70%, respectively (Figure 6), whereas the pass rates of its middle-class counterpart are 88.73%, 90.79%, and 86.36%, respectively (Figure 6). The pass rates for lower-class schools in the same years are 85.70%, 88.93%, and 84.82%, respectively (Figure 6).

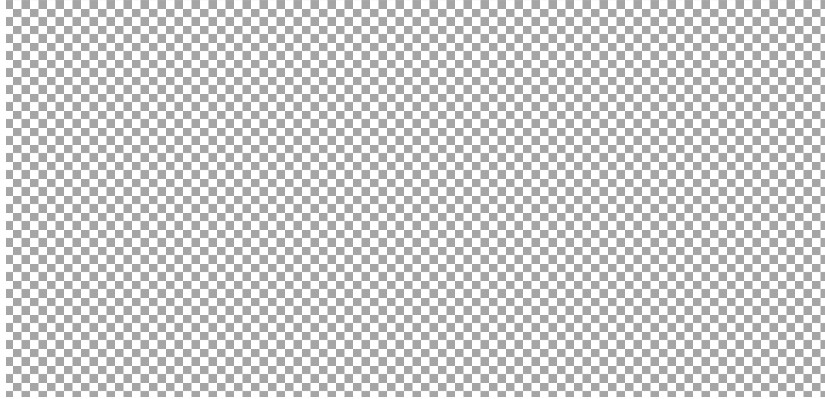

**Figure 5.** *Cluster wise overall pass rate in different years.* Horizontal axis represents clusters, and vertical axis represents cluster-wise percentage share of performance of students.

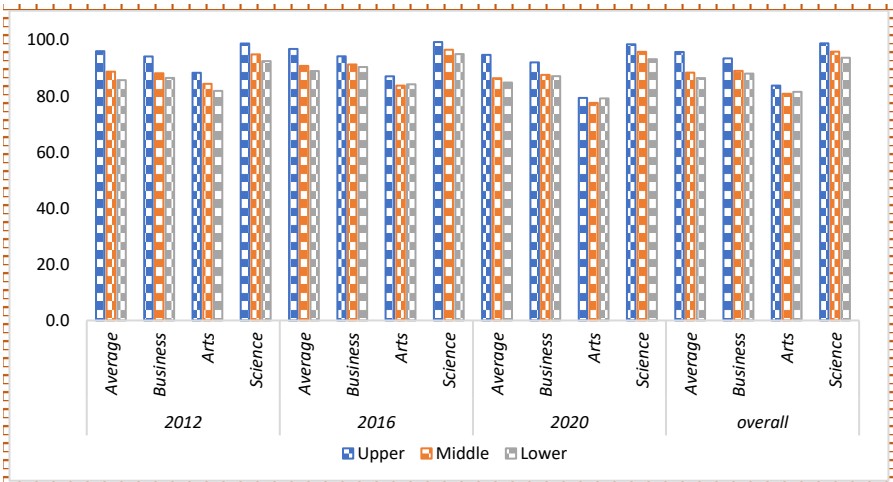

**Figure 6.** *Pass rate of students with different economic classes and different clusters.* Horizontal axis represents cluster and economic status, and vertical axis represents percentage share of performance of students of three economic classes and clusters.

The pass rate in upper-class schools is exceptionally better, and the percentage of GPA 5 holders in the upper class is extremely high. For instance, the percentage of GPA 5 holders for upper-class schools in the years of 2012, 2016, and 2020 were 44%, 44.40% and 40.29%, respectively (Figure 7), whereas GPA 5 holders for their middle-class counterparts belonged to 11.01%, 10.52%, and 10.35% of students, respectively (Figure 7). The GPA 5 holders for lower-class schools in the same years were 3.85%, 4.69%, and 4.88%, respectively (Figure 7). In the competition of academic success, neither middle nor lower-class schools could compare with their upper-class counterparts, which, in education, are home to elites. Now, the performance of the three clusters, namely science, business, and the arts, is compared.

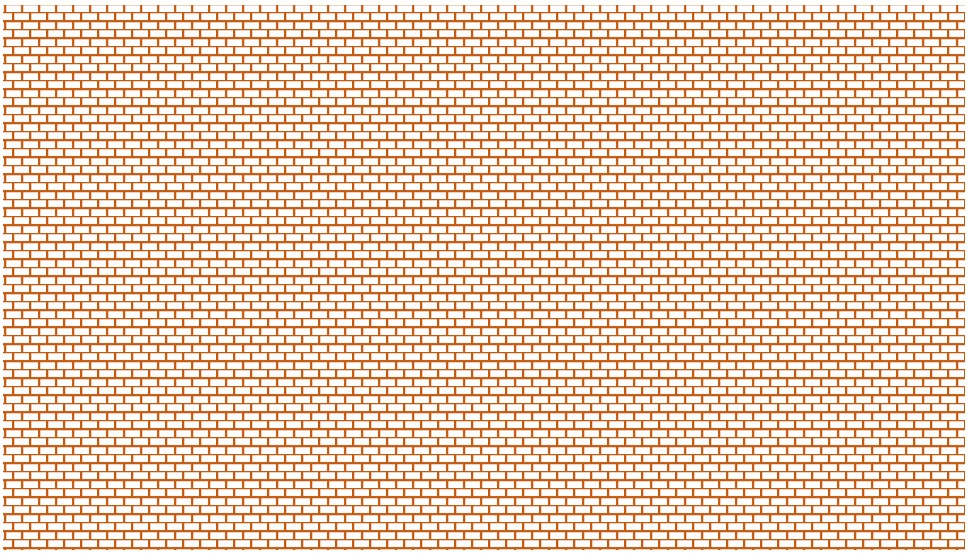

**Figure 7.** *Cluster-wise students with GPA 5 from different economic classes.* Horizontal axis represents cluster, and vertical axis represents percentage share of three clusters.

The pass rates for science in the years of 2012, 2016, and 2020, regardless of school status (upper, middle and lower class) were 95.07%, 96.71%, and 95.65%, respectively (Figure 8), whereas those of business were 88.07%, 91.19%, and 87.89%, respectively (Figure 8). The pass rate for the arts in the same years were 82.98%, 84.13%, and 78.50%, respectively (Figure 8). Although the pass rate for the science cluster was exceptionally higher, the percentage of GPA 5 holders in the science cluster was substantially divergent. For instance, the GPA 5 holders for science in the years of 2012, 2016, and 2020 were 34.88%, 52.10%, and 32.83%, respectively (Figure 8), whereas the GPA 5 holders of the business counterpart were 5.71%, 2.50%, and 1.88% (Figure 8). The GPA 5 holders for the arts in the same years were 0.85%, 0.34%, and 0.61%, respectively (Figure 8).

This simply suggests that both the business and arts clusters were unable to compete with the science counterpart in academic competition. Hence, SES has greatly influenced academic performance, in which elites succeed the most. Science education and having a better performance within this cluster have become an exclusive phenomenon for elites. The following sub-section explains the impact of elitism on science education.

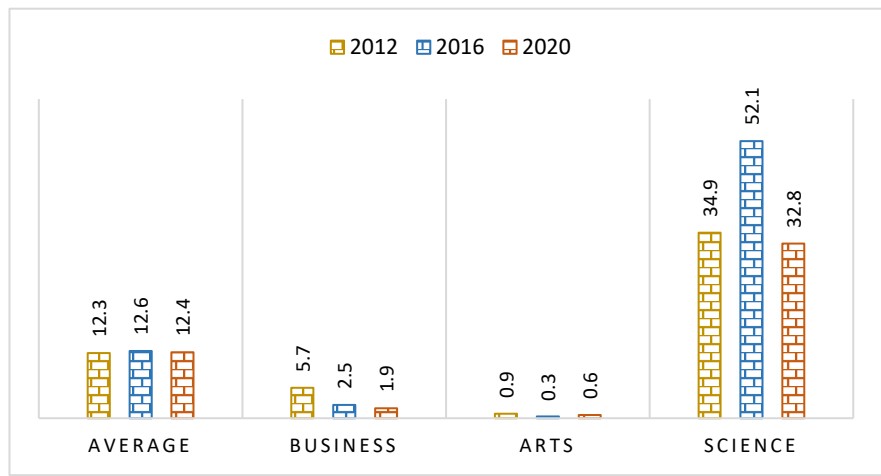

**Figure 8.** *Distribution of GPA 5 holders amongst three clusters in respective three years.* Horizontal axis represents clusters, and vertical axis represents percentage share of GPA 5 holders.

### 5.3. Reason for Elitism in Science Education and its Impact

Two root causes have been found that make science education an elite product, and the other reasons are branches. The following discussion highlights the identified two root causes. Firstly, education is no longer an agent of change making in the 21st century, as confirmed by all teachers, parents, and students involved in the FGDs. According to TSCU, TSCM, TSCL, TBUSU, and TBUM, education has rather become an agent that helps us to adopt changes that have already taken place internationally, especially in the West. Thus, injecting the capacities and skills required for change making is not a prime concern for education (PSCU, PSCM, TSCU, TSCM, and TSCL). Hence, education should rather inject the skills and capacities that help students to adopt changes that have already taken place (TSCU, TSCM, and TSCL).

Science education is the only program that is able to inject such skills needed to adopt the changes that have already taken place, especially in the West (TSCU, TSCM, and TSCL). Hence, science has become an international education program. The development of course curricula, instructional technology, quality assurance, and assessments of science education are developed internationally, especially in the West, which is then subsequently exported to its eastern counterpart (PSCM, PSCU, TSCU, TSCM, and TSCL). As an international product, the learning objectives and outcomes of science education are very compact, as suggested by the international theory. Hence, students need to learn theoretical concepts of science rigorously without much adherence towards practical experiments required for a different context.

Under the mentioned circumstance, indicating a theoretical answer accurately without checking contextual fitting would ideally ensure a full score in the assessment of science education. On the other hand, a minor mistake would not impact the score. Hence, the theory of science needs to be captured carefully. In doing so, support from schools is neither conducive nor exclusive, as suggested by all teachers, parents, and students. Students are required to take additional support via private coaching, as noted by all students, parents, and teachers. Teachers are allowed to offer private coaching, which consumes extra funds from students and parents. It was found that all teachers teaching science programs offered private coaching. A teacher commented the following:

> *"I am proud to be a science teacher as learning and teaching science are highly prestigious in our society. On the other hand, although teacher teaching subjects related to science programs is not offered a different salary package in the school compared to a teacher teaching subjects related to business and arts programs; by offering private coaching, we can earn at least 10 times more compared to the colleagues who are teaching subjects related to arts and business programs. Science is the only hope and only promise that we need to adapt to cope in the 21st century".*

Although these comments sound promising from an individual perspective, they indicate a great concern. Hence, students from privileged SESs can use such a scope to learn science. On the other hand, although business is a semi-international product, the arts is a local product, as said by all teachers. Although it is not possible to achieve a full score in these areas, attempts by individual students may help to obtain scores to pass the programs, as expressed by the SATU, SATM, SATL, SBUM, and SBUL. With this hope, many students with lower SESs study the arts and business programs at least to have some form of qualification before they depart from the education system.

Secondly, the policy perspective has also made science education an elitist product. According to the policy, students from science can pursue further education in any area, such as science, business, social science, and the arts, whereas their counterparts from business and the arts must follow the same path for higher education, continuing to study what they have studied in the secondary provision. Score achieved in the SSC is an important prerequisite to continue with further education.

A greater score achieved in the science program always accelerates the chances for science students to continue with further education in one of the possible clusters, namely science, business, and the arts. Thus, many students from the arts and business clusters leave the education system without pursuing further education. Hence, many secondary science graduates continue with further education in business, the arts, and social sciences. Can patronising only elites in education be a goal for sustainable development? This is a wake-up call for international education and development research. The following writing reports the impact of elitism in science education.

In addition to several causal effects, two central adverse impacts that have developed as a result of elitism in science education are the subjects of the following. Firstly, the misuse of public funds in the name of education is a serious concern. Most students, especially those who study business and arts programs, discontinue education after the secondary provision and subsequently join the labour force (SBUL, SBUU, PBUM, PBUL, SATU, SATM, SATL, TBUM, TATM, and TATL). It was further reported that these students drop from the education system without accruing any form of low or semi-skills needed for their employment.

From conversations with students, parents, and teachers, it is understood that, after leaving the education system, the assignments that adolescents from the business and arts clusters undertake to survive require a substantial amount of knowledge in applied science. Unfortunately, the education system does not inject such knowledge. Hence, public funds and the time spent educating in business and the arts may significantly be misused.

On the other hand, the continuity of further education in business, the arts, and social science by secondary science graduates is a complete misuse of both financial and logistic resources. Moreover, the occupancy of employment in the area of social science, business, and the arts by a science graduate develops both horizontal and vertical mismatch [58–60]. This is what is called either a waste of resources and time or corruption via policy. Should the emotions and time of youths who are from lower SESs be played with in the name of elitist views in education? This is an emerging perspective on account of corruption and crime via legitimate policies, as argued by [1,6].

Secondly, the elitism view in education has challenged the philosophy of education contributed by Aristotle, Plato, Socrates, and Ibn Sina. The fundamental philosophy of primary education is to inject citizenship skills and values, whereas secondary school should add to the skills and competencies needed to undertake low and semi-skilled jobs [6]. In doing so, both the primary and secondary provisions are cautioned to inject skills and values needed to continue higher education for a very selective portion of students who are extraordinarily capable, both in theory and practice.

The purpose of the education system is not to develop a specific group as an extraordinarily capable counterpart. The system should rather support the natural extraordinarily capable group regardless of SES. This is an idealist view that the current society hopes to achieve one day in the education system but that, unfortunately, it has yet to have.

Time is needed for transformation towards an idealistic view in education that is needed for sustainable development. However, it is not acceptable to welcome purposeful discrimination or discrimination developed diplomatically in the era of MDGs and SDGs. The following discussions suggests how to develop a mechanism that may support the substantial development of education.

*5.4. Elitism in Education—A Mechanism for Sustainable Education*

Elitism in education limits the sustainable development of education not only by bringing disparity but also by dismantling both the philosophy of education and the reciprocal relationship that has prevailed amongst different provisions of education (such as primary, secondary, and tertiary education). The following proposed mechanism for the sustainable development of education is not a new phenomenon; it is, rather, a re-stated fact that emerging nations, including Bangladesh, need to pay serious attention to.

Firstly, the purpose of education and the role of the specific provision of education (such as universal, secondary, higher secondary, and higher education) need to be redefined by following the philosophy of education without allowing the market to rule. In the process of redefining the purpose of education and the role of a specific provision, a country should not simply follow international trends, but it should discover a way that it can compete in the global race of education [6,28,34,35].

Secondly, a country should ensure unified universal education for all school-aged children [41–43]. Universal education, in particular primary education, should not simply be a supplier of graduates for its secondary counterpart. The success of primary education should not be measured on the basis of its capacity to produce competent graduates to study secondary education [41–43]. The success of the primary provision should be judged based on its competency to provide citizenship skills to all its graduates regardless of SES. Hence, stakeholders and society should be the main assessors of primary graduates, not just the teachers who are teaching them [5,7,47,48].

Based on academic ability and interest, a selective portion of universal graduates, regardless of SES, should travel for further education. Secondary education should offer an open choice of subjects for all its students. Students should be able to consider several subjects from different clusters that they have interest and competence in. Higher education must be based on academic ability, competency, and interest without giving any form of prejudice to SES. In the event of furthering education, students need to study prerequisite subjects in the same area in earlier provisions.

If anyone wishes to study a particular subject in higher education that he/she did not learn in earlier provision(s), he/she should re-take the relevant subject(s) from the secondary provision before being admitted to higher education. Students who wish to discontinue education after secondary education need to have some form of education and training by covering either trade, another sector, or both before leaving the education system, which might be helpful for one's potential career path in the future. Hence, information management on demography, sector of employment, and their required skills (such as type of skills and level of skills) are important.

Planning in higher education must avoid elitist views. It also needs to avoid a tendency to be a free market for all secondary graduates. A selective portion of capable secondary graduates, regardless of their SES, should study higher education. Higher education should not meet the demand of "diploma disease". Enrolment in higher education and its expansion should not be based on the supply of secondary graduates; rather, it needs to focus on the needs of the employment market. The qualification levels and their diversifications are controlled by the needs of labour market and its skill requirements, not just by the demands created as a result of graduates produced in earlier provisions (such as secondary, higher secondary, and bachelor programs).

## 6. Implications, Further Research, and Limitations

The following includes two major theoretical and practical implications before outlining topics for further research and acknowledging limitations.

### 6.1. Theoretical Implications and Practice

The disparity and dependency theories and their impact on educational development are often used in conceptual connotations [1,2,6,28]. Although the first theory refers to a scenario of discrimination experienced onshore [6], the latter is used to explain an offshore scenario [6,28]. Hence, this is the first piece of research that explores the connection between the disparity and dependency theories. The findings and discussions suggest that both theories can supplement and complement each other and can also further aggravate the situation. Hence, the dependency theory, referred to in this paper as the internationalisation of science education, is supporting the development of the disparity theory, referred to as elitism in education. This is a new theoretical phenomenon established in this study that adds value to both the disparity and dependency theories.

The philosophy of education identified in earlier centuries has been the key to making education a product of ideologist and novelist views [7,15,26], which have subsequently helped education become popular and acceptable within society [6,15]. Hence, education has become a social product, and everyone within society contributes to and fosters its development [6,28,29]. Taking advantage of the popularity of socialist views in education, a capitalist view is silently practiced [6,28,29]. The silent practice of capitalist views is slowly corrupting the philosophy of education [34,39]. Hence, further revised practice guidelines are provided to confront the challenges involved in the education of the 21st century, in which the original philosophy of education is retained—a major practical contribution of this research. Before using the suggested guidelines, more scenarios in this field should be explored by conducting further studies, as suggested below.

### 6.2. Further Research and Limitations

The findings and discussions suggest that private investment in education (such as private coaching and tuition, private and elite schools, and many forms of informal funds) is developing a capitalist view in science education, which subsequently supports science education as an elitist product. Funds have always been an important issue, especially for emerging nations. Hence, neglecting private investment may add further problems. Therefore, new research focused on discovering a better model using private investment without waiving the philosophy of education is would be beneficial.

The findings and discussions suggest that a significant portion of students leave the education system after the secondary provision. Injecting employable semi-skills before their departure is important. It is thus important to map the employment sectors that students engage with after departing secondary education and their skill dynamics. This can help reshape secondary education for school leavers, following the philosophy of education. Hence, topics that cover mapping skill requirements for secondary school leavers would be good topics to explore in future research.

Gender perspectives, which are significant for STEM studies, were not considered in this study because an exclusive study on gender in STEM would provide more insight instead of making a sweeping argument; a number of studies have already covered gender issues in STEM; and finally, information/data were collected based on students' SSC IDs numbers from public archive(s), which did not allow tracing student genders, thus preventing the gender issue from being explored.

This is a small-scale research work that does not allow for comparison with many nations. Henceforth, more protracted longitudinal research backed up by the sponsorship of governments and development partners on these issues and other issues suggested for further research would be helpful. Extensive research should yield important insight not discovered here. Such insight would benefit the education system in the eastern part of the globe.

## 7. Conclusions

The role of science education on development in the era of SDGs and post MDGs is more than inexorable. However, elitism in science education has developed a disparate atmosphere within the education system in the name of internalisation. This situation has further created hindrances for sustainable development in education—a prerequisite for national development in the era of SDGs. Education that lives in isolation from its philosophy and purpose may provide temporary assistance towards national development, either by following capitalist views or by developing a purposeful role. This is a short-sighted view in educational development and may be a problem that hinders the sustainable development of education. Urgent revision of the purposes and roles of national education and their decent implementation are badly needed in light of the philosophy of education—an appeal made in this study.

**Funding:** This research received no external funding.

**Institutional Review Board Statement:** Not applicable.

**Informed Consent Statement:** Not applicable.

**Data Availability Statement:** Data are collected from the public archives named Web-based Results Publication System for Examination Boards (WBRSEB) which can be accessed from https://eboardresults.com/v2/home.

**Acknowledgments:** I would like to express my sincere thanks to my doctoral students namely Zhou Lei and Mahfuzur Rahman for their kind support to read this work and to provide me the feedback. Thanks to Syed Abu Mazher for his support in the illustration of the Bangladesh map that is used. I also owe my heartfelt gratitude to Morsheda Parvin and without her support, it was impossible to collect the large amount of data and organised them from the public archives.

**Conflicts of Interest:** The authors declare no conflict of interest.

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
