# Peer review of "Has Secondary Science Education Become an Elite Product in Emerging Nations?—A Perspective of Sustainable Education in the Era of MDGs and SDGs"

_sustainability, doi:10.3390/su15021596_

Round 1
Reviewer 1 Report
This paper addresses a topic of great interest and actuality - access to a quality education for all students and the role of socio-economic conditions in the educational process.
The strong point of the article is the broad and well-documented approach to the chapters regarding the concept of development, the relationship between development and education and the context of research.
The quality of the paper would increase if the tables (table 1, 2, 3, 4) were inserted into the text and not attached to the end of the article. As well as figures 1 - 7.
Also, regarding Table 1 - Number of school by locations and their status - I believe that it should be replaced by a cartogram, using the map of Bangladesh at the district level, the number of schools in each district being represented with a dimensioned circle. A map with the territorial distribution of schools would be much more expressive than a table.
Please pay attention to the numbering of chapters and subchapters. If there are 2.1 and 2.2, then why is there 1.1 (1.1. Relationship between development and education)?
Finally, I would like to note that the author cites no less than 12 of his own articles (as first author), which can be interpreted as excessive self-citation.
Author Response
The Editor  
Sustainability
Subject: revised version submission for (sustainability-2130043-R1)
Dear Editor
I would like to thank you for getting our manuscript reviewed. Our heartfelt gratitude towards the reviewers for their meticulous jobs and without which it would not have been possible to reach this stage. We are indebted to you for your professional guidance.
The following is a write-up which may kindly be accepted as a response to the comments from the reviewers and from the editors.
General Responses:
I have found all the comments are important and incorporated the corrections suggested. While the corrections given by reviewer 1 are related to technical correction, reviewer 2 suggested some corrections on the writing. Although I respect the comments from all reviewers and put all the efforts to address them. The corrections are made by the blue colour texts and they are found in the revised version.
Having saying that a point-by-point detailed explanation on how I have addressed each comment is provided bellow.
---------------------------------------------------------------------------------------------------------------------
Response to Reviewer 1
Comment:
This paper addresses a topic of great interest and actuality - access to a quality education for all students and the role of socio-economic conditions in the educational process.
The strong point of the article is the broad and well-documented approach to the chapters regarding the concept of development, the relationship between development and education and the context of research.
Response:
Thank you so much. I appreciate your time and valuable comments on our paper. Your comments are substantially crucial to our study.
Comment:
The quality of the paper would increase if the tables (table 1, 2, 3, 4) were inserted into the text and not attached to the end of the article. As well as figures 1 - 7.
Response:
I am sorry that I missed to inset where the table, figures and diagram are to be set in the texts. I have now inserted that note to direct where the table, figures and diagram are to be set and please be assured that they are not annex. However correctly, they can be found at the bottom of the paper but typesetting would make sure that they are inserted at the right place while final formatting would take place.
Also, regarding Table 1 - Number of school by locations and their status - I believe that it should be replaced by a cartogram, using the map of Bangladesh at the district level, the number of schools in each district being represented with a dimensioned circle. A map with the territorial distribution of schools would be much more expressive than a table.
Response:
Thanks for the great suggestion. Following your suggestion, a map is also incorporated.
Please pay attention to the numbering of chapters and subchapters. If there are 2.1 and 2.2, then why is there 1.1 (1.1. Relationship between development and education)?
Response:
Thanks for the great suggestion. Due care is taken to address this issue.
Finally, I would like to note that the author cites no less than 12 of his own articles (as first author), which can be interpreted as excessive self-citation.
Response:
Thanks for the great suggestion. Actually, not much works has conducted to back the statement thus have to rely on self-citation. Having said that 15% self-citation is under tolerance as far as international policy since these studies are receiving wider citation from global community, we believe that we are under tolerance level.
I sincerely thank for the support and I am obliged. I made all the corrections as suggested therefore pray and hope the esteemed reviewers and editor grant the publication.
Yours sincerely
Author

Reviewer 2 Report
I think it is a very interesting study but some aspects need to be carefully adressed, in my point of view:
FGDs was used in the abstract with no previous explanation
The approach to EFA, MDG and SDG does not inlcude the human right perspective of education that inspires all of them, specially the last one, and the principle of non disrimination that is the mandatory one for this right since UNESCO Convention (1960). I think it is very important to address the state of ratification and report of the country with this international treaties to understand the context and relation with the evolution of such agendas.
Limitations of the study were not included in the text and some of them concerning this human rights approach to education should be addressed, e.g. STEM and women/girl discrimination to education or some other vulnerable groups of student that suffer greater SES.
Gender perspective should be included in the research questions.
Author Response
The Editor  
Sustainability
Subject: revised version submission for (sustainability-2130043-R1)
Dear Editor
I would like to thank you for getting our manuscript reviewed. Our heartfelt gratitude towards the reviewers for their meticulous jobs and without which it would not have been possible to reach this stage. We are indebted to you for your professional guidance.
The following is a write-up which may kindly be accepted as a response to the comments from the reviewers and from the editors.
General Responses:
I have found all the comments are important and incorporated the corrections suggested. While the corrections given by reviewer 1 are related to technical correction, reviewer 2 suggested some corrections on the writing. Although I respect the comments from all reviewers and put all the efforts to address them. The corrections are made by the blue colour texts and they are found in the revised version.
Having saying that a point-by-point detailed explanation on how I have addressed each comment is provided bellow.
---------------------------------------------------------------------------------------------------------------------
Response to Reviewer 2
Comments:
I think it is a very interesting study but some aspects need to be carefully adressed, in my point of view
Response:
Thank you so much. I appreciate your time and valuable comments on our paper. Your comments are substantially crucial to our study. I address all of your comments in the revised version. The point-by-point following note describes how I address your comments.
Comments:
FGDs was used in the abstract with no previous explanation
Response:
The explanation is made.
Comments:
The approach to EFA, MDG and SDG does not inlcude the human right perspective of education that inspires all of them, specially the last one, and the principle of non disrimination that is the mandatory one for this right since UNESCO Convention (1960). I think it is very important to address the state of ratification and report of the country with this international treaties to understand the context and relation with the evolution of such agendas.
Response:
Thanks. Following your suggestion, we have developed a new sub-section at 2.4 entitled Convention against discrimination in education 1960 and interventions: flashback where your points are well taken care off. Thank you so much for the kind suggestion which is very pragmatic. This can be found in page 8 with blue colour texts.
Comments:
Limitations of the study were not included in the text and some of them concerning this human rights approach to education should be addressed, e.g. STEM and women/girl discrimination to education or some other vulnerable groups of student that suffer greater SES.
Response:
We have added that and they can be found in pages 23 and 24 with blue colour texts.
Comment:
Gender perspective should be included in the research questions.
Response:
The reason that has restricted us not to include gender issue is explained under limitation with blue colour texts at pages 23 and 24. Also, further reason is provided at page 3 with blue colour texts.
In addition to that in response to your suggestion on English editing, we have taken necessary action and further contract is made with Mr. Wohler Huang (Section Managing Editor: responsible for this SI). He suggested if required, the MDPI will arrange editing.
I sincerely thank for the support and I am obliged. I made all the corrections as suggested therefore pray and hope the esteemed reviewers and editor grant the publication.
Yours sincerely
Author

Round 2
Reviewer 2 Report
All the remarked issues were addressed.